# Oxidation of Thiosulfate with Oxygen Using Copper (II) as a Catalyst

**Juan Manuel González Lara [1], Francisco Patiño Cardona [2], Antonio Roca Vallmajor [3],\* and Montserrat Cruells Cadevall [3]**

[1] Departamento de Biotecnología, Universidad Politécnica de Pachuca, Carr. Pachuca-Ciudad Sagahún km 20, Ex-Hacienda de Santa Bárbara, Zampoala Hidalgo 43830, Mexico; joanmanelgl@yahoo.es

[2] Ingeniería en Energía, Universidad Metropolitana de Hidalgo, Boulevard acceso a Tolcayuca 1009, Ex-Hacienda San Javier 43860, Mexico; franciscopatinocardona@gmail.com

[3] Departament de Ciència de Materials i Química Física, Facultat de Química, Universitat de Barcelona, Martí i Franquès 1, 08028 Barcelona, Spain; mcruells@ub.edu

\* Correspondence: roca@ub.edu; Tel.: +34-934021295

**Abstract:** Thiosulfate effluents are generated in the photography and radiography industrial sectors, and in a plant in which thiosulfates are used to recover the gold and silver contained in ores. Similar effluents also containing thiosulfate are those generated from the petrochemical, pharmaceutical and pigment sectors. In the future, the amounts of these effluents may increase, particularly if the cyanides used in the extraction of gold and silver from ores are substituted by thiosulfates, or if the same happens to electronic scrap or in metallic coating processes. This paper reports a study of the oxidation of thiosulfate, with oxygen using copper (II) as a catalyst, at a pH between 4 and 5. The basic idea is to avoid the formation of tetrathionate and polythionate, transforming the thiosulfate into sulfate. The nature of the reaction and a kinetic study of thiosulfate transformation, by reaction with oxygen and $Cu^{2+}$ at a ppm level, are determined and reported. The best conditions were obtained at 60 °C, pH 5, with an initial concentration of copper of 53 ppm and an oxygen pressure of 1 atm. Under these conditions, the thiosulfate concentration was reduced from 1 g·L$^{-1}$ to less than 20 ppm in less than three hours.

**Keywords:** thiosulfate oxidation; kinetics; catalysis; oxygen

## 1. Introduction

In metallurgy, there is growing interest in technology for the extraction of gold and silver that does not use cyanide, for application in minerals or in residues in which it is difficult to recover metals from leaching liquids, and also for environmental purposes. The dissolution of gold and silver by thiosulfate occurs via the formation of the corresponding metal complexes. Technological proposals for implementing this process include the use of a copper (II) salt as an oxidizing agent and ammonia for copper stabilization within the system. One advantage in the use of thiosulfates is the reduction of interaction with other cations, such as copper, arsenic, and antimony, in contrast to the case of cyanidation. The thiosulfate route for the recovery of gold and silver from ores using Cu (II) and ammonia is complex. The leaching rates are acceptable, but the consumption of reagents can be high due to thiosulfate degradation. There are also difficulties encountered during the recovery of the dissolved metals [1–11]. Several authors have proposed modifications of the process reported in the literature. An alternative process includes ferric-ethylenediaminetetraacetic acid (EDTA), ferric-oxalate, and ferric-citrate to avoid the difficulties with the cupric-tetra-amine additive, including excessive thiosulfate oxidation and high reagent costs [12], whereas other authors presented an analysis of

the effect of EDTA, thiosulfate, and cupric ions on silver leaching kinetics [13]. At present, however, thiosulfate leaching of minerals is only applied commercially in the Goldstrike deposit of the Barrick Gold Corporation [8].

Meanwhile, every year, millions of tons of waste are generated worldwide from electrical and electronic products. These devices are an important source of secondary raw materials in the form of gold and other metals with high economic values [14,15]. The amount of gold on the printed circuit boards (PCBs) of mobile phones can reach 300–350 g·ton$^{-1}$ [16]. A leaching process using thiosulfate in an ammoniacal medium to recover the gold contained in these PCBs has also been reported [17]. The authors indicated that the leaching system offers promising opportunities for industrial application; moreover, optimization of the leaching system for recovering gold from such PCBs was also carried out [14].

One of the most important uses of thiosulfate solutions has been for application in the photographic industry (in fixing baths and others) to dissolve the silver halide not reduced to metallic silver during photographic or radiographic processes. Today, the industry still generates significant amounts of thiosulfate-based effluents that need to be detoxified for environmental purposes, and to recover the silver [18–21].

In recent decades, for gold and silver coatings, the sulfite-thiosulfate system has been proposed as a non-cyanide coating technology [22–27].

As well as the thiosulfate effluents generated in the photography and radiography industries, the plants that use thiosulfates to recover the gold and silver contained in ores, and also the petrochemical, pharmaceutical, and pigment industries, among others, all produce thiosulfate effluents. In the future, the amounts of these effluents may increase considerably, particularly if cyanides are substituted by thiosulfates in the extraction of gold and silver from their ores, in the recovery of metals contained in electronic scrap, or in metal coating processes. Therefore, an adequate process is needed to degrade the thiosulfate contained in the corresponding effluents.

Different processes for this degradation of thiosulfates have been proposed. A study of the oxidation of thiosulfates by oxygen, using synthetic sphalerite doped with transition metals [28], the oxidation of thiosulfate with $H_2O_2$ in a catalyzed reaction, being the oxidation products sulfite and sulfate [29], and the oxidation of thiosulfates contained in wastewater, with oxygen in the presence of UV light [30], have been carried out.

Our research team studied the transformation of thiosulfate using copper sulfate solutions. The amount of copper (II) salt used was almost the stoichiometric quantity for the formation of 1 mol CuS and 1 mol sulfates per mol thiosulfate. The transformation of thiosulfate using copper (II) sulfate was applied to an industrial fixing bath from the photographic industry; the final effluent contained less than 10 mg L$^{-1}$ of thiosulfate [31].

This paper reports a study of the oxidation of thiosulfate to sulfate, with oxygen using copper (II) as a catalyst. The reaction was developed at a pH between 4 and 5 and the idea is to avoid the formation of tetrationate or polythionate. The nature of the reaction and its kinetics, with oxygen and in this case with Cu (II) at the ppm level, are determined. The effects of temperature and the initial concentrations of $H_3O^+$, $Cu^{2+}$, and $S_2O_3^{2-}$ on the thiosulfate transformation are also determined; and the effect of the partial pressure of oxygen is studied.

## 2. Materials and Methods

### 2.1. Materials

In the study of thiosulfate degradation, sodium thiosulfate solutions prepared from this salt, with 99% purity, and pentahydrate copper sulfate of the same purity were used. Oxygen with a purity of ≥99.995% (mol/mol) was also used.

*2.2. Experimental Procedure for Thiosulfate Transformation Using Copper Sulfate*

The experiments were carried out in a 0.5 L flat-bottom temperature-controlled reactor with magnetic stirring. The pH was kept constant by adding the necessary amounts of 0.5 mol·L$^{-1}$ H$_2$SO$_4$ solution and 0.5 mol·L$^{-1}$ NaOH solution.

In each experiment, the stirring rate, pH, air/oxygen flow, and temperature were adjusted to the values required in each case. Samples were taken at different times and filtered; and the remaining thiosulfate in the liquids was analyzed by iodometry (iodine 0.1 mol·L$^{-1}$ was used for this purpose and a starch solution was added to determine the end point). Some samples were also analyzed via ionic chromatography. During the process, the solids generated were characterized by X-ray diffraction (XRD, PANanalytical, Almelo, The Netherlands) as well as by scanning electron microscopy (SEM, JEOL, Tokyo, Japan), and energy-dispersive X-ray analysis (EDS, Oxford Instruments, Abingdon, England). The rates of these processes are indicated from the $k_{exp}$ values obtained in each experiment (slope of the graph: conversion, *X*, versus time); the conversion was defined according to the expression:

$$X = [S_2O_3{}^{2-}]_{transformed} / [S_2O_3{}^{2-}]_{initial.} \tag{1}$$

## 3. Results and Discussion

Preliminary experiments involving thiosulfate oxidation by oxygen, but without the presence of Cu (II), led to very slow rates for the process.

*3.1. The Nature of the Thiosulfate Transformation by Reaction with Cu (II) Sulfate and Oxygen*

Two experiments involving thiosulfate degradation using Cu (II) were carried out under the following conditions: temperature, 60 °C; stirring rate, 500 min$^{-1}$; initial concentration of thiosulfate, 1 g·L$^{-1}$; initial concentration of Cu$^{2+}$, 8.4 × 10$^{-4}$ mol·L$^{-1}$ (0.053 g·L$^{-1}$); and oxygen pressure, 1 atm. The first experiment was carried out at pH 5, and the second at pH 4.

Figure 1 shows the results for thiosulfate transformation at the indicated pH values. The thiosulfate degradation was almost total for a reaction time of 160 min. Meanwhile, two different rates were obtained: a slower one during the first part of the process, and a faster one during the second part. The values obtained were similar in both cases, pH 5 and pH 4. At a conversion of nearly to 0.40 (both experiments), the change in the slope was observed as the reaction rate increased, and a black solid was formed: CuS as confirmed by XRD (Figure 2). Therefore, it seems that the presence of this CuS precipitate enhanced the reaction rate, indicating a catalytic effect on thiosulfate degradation of the contact with copper sulfide surface. The precipitate of copper sulfide consisted of aggregates of crystals with 1 μm size (SEM-EDS).

Final liquids were analyzed by ion chromatography, indicating that thiosulfates were oxidized to sulfates under the experimental conditions employed (see Figure 3). In the pH interval between 4 and 5, sulfates were detected; any other species, such as tetrathionate or polytionate, were not detected.

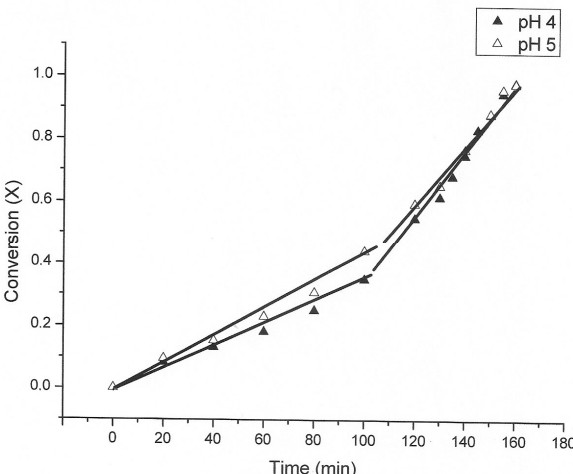

**Figure 1.** Nature of the reaction for thiosulfate transformation using $Cu^{2+}$ and oxygen: $[S_2O_3{}^{2-}]_{initial}$, $1\ g \cdot L^{-1}$; $[Cu^{2+}]_{initial}$, $0.053\ g \cdot L^{-1}$; $pH_{constant}$, 4 or 5; 60 °C.

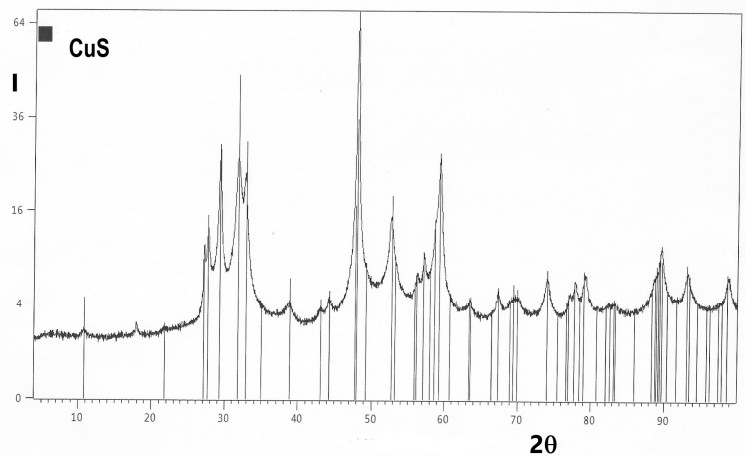

**Figure 2.** Nature of the reaction: X-ray diffractogram of solid obtained during thiosulfate transformation: identified as CuS.

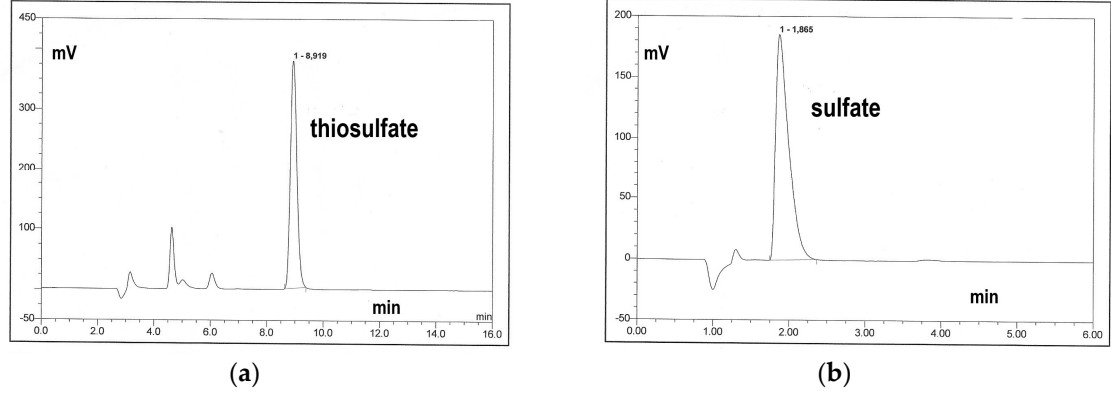

**Figure 3.** Chromatograms: (**a**) Initial solution; (**b**) Final solution.

When CuS had formed (second step), hydronium ions were generated and pH was kept constant by the addition of sodium hydroxide solution.

The amount of thiosulfate remaining in the solution at the end of the experiment was less than $20\ mg \cdot L^{-1}$ in both cases. Table 1 includes the slopes of the graphs $X_{thiosulfate}$ vs. time ($k_{exp}$), which is a measure of the transformation rate.

**Table 1.** Values of $k_{exp}$ of thiosulfate transformation in two steps.

| pH | Time (min) First Step | Time (min) Second Step | $k_{exp}$ (min$^{-1}$) First Step | $k_{exp}$ (min$^{-1}$) Second Step |
|---|---|---|---|---|
| 5 | 0–80 | 80–160 | 0.0040 | 0.0070 |
| 4 | 0–100 | 100–160 | 0.0033 | 0.0110 |

Iodometry established the absence of sulfites, while tetrathionates or polythionates were not detected by chromatography. Thus, the initial thiosulfates were transformed to sulfates.

Under the employed experimental conditions, the reaction produced in the zone where the change in the slope occurs could be as follows:

$$S_2O_3{}^{2-}{}_{(aq)} + Cu^{2+}{}_{(aq)} + 3H_2O \Rightarrow SO_4{}^{2-}{}_{(aq)} + CuS_{(s)} + 2H_3O^+{}_{(aq)}. \tag{2}$$

This process is quite similar to the degradation process using large amounts of copper ions (0.21–0.85 g·L$^{-1}$ Cu). In that case, sulfates and copper sulfide were formed in a similar amount in mol, when >0.57 g·L$^{-1}$ Cu were used. After this, the copper sulfide formed, and had to be oxidized to copper sulfate prior to being reused in the next cycle [31]. By using 0.053 g·L$^{-1}$ Cu (in the present work) and oxygen, the reaction rates are slower, but the transformation of thiosulfate to sulfate takes place in more than 90% yield. In this case, the amount of copper sulfide that must be oxidized to sulfate is much lower than the work in which large amounts of copper were used.

Another difference between these two procedures is that when we used very low amounts of copper salts and oxygen, two rates were detected; whereas, when using larger amounts of copper (without oxygen), only one rate value ($k_{exp}$ = 0.013 min$^{-1}$) was determined, and the thiosulfate degradation was completed in a shorter time. The work cited [31] also demonstrated that, in experiments carried out at pH 6 and pH 10, no thiosulfate transformation to sulfate was detected. Meanwhile, thiosulfates were easily decomposed at pH $\leq$ 4, giving HSO$_3{}^-$, SO$_2$, and elemental sulfur. For these reasons, the kinetic study in this work was carried out in the pH interval between 4 and 5.

### 3.2. Kinetic Study of the Transformation of Thiosulfate by Reaction with Oxygen and Cu$^{2+}$

The effects of temperature (40–80 °C), concentration of H$_3$O$^+$ (1.0 × 10$^{-4}$–1.0 × 10$^{-5}$ mol·L$^{-1}$), initial Cu$^{2+}$ (4.2 × 10$^{-4}$–1.68 × 10$^{-3}$ mol·L$^{-1}$), and initial S$_2$O$_3{}^{2-}$ (4.5 × 10$^{-3}$–9 × 10$^{-3}$ mol·L$^{-1}$), as well as an oxygen pressure of 0.2–1 atm, were determined.

Figure 4 shows the plot of conversion vs. time at different temperatures. Table 2 includes the experimental values of $k_{exp}$ for these experiments; the table includes the corresponding values corrected by the oxygen concentration at each temperature, because the oxygen concentration varies with temperature [32]. The activation energy was calculated according to the following expression:

$$(k_{exp}/[O_2]) = Aexp(E_a/RT). \tag{3}$$

Figure 5 includes the plot of ln($k_{exp}/[O_2]$) vs. 1000/$T$. Two apparent activation energies E$_a$ were obtained: for the first step, 41 kJ·mol$^{-1}$ and for the second, 33 kJ·mol$^{-1}$.

For concentrations of H$_3$O$^+$ varying from 1.0 × 10$^{-4}$ to 1.0 × 10$^{-5}$ mol·L$^{-1}$ H$_3$O$^+$ (pH between 4 and 5, Figure 6), the reaction rate, $k_{exp}$, in the first step was between 0.0033 and 0.040 min$^{-1}$; and, in the second step, it was between 0.070 and 0.0110. All the values obtained are similar; consequently, an apparent reaction order of ≈0 with respect to the hydronium concentration was obtained.

The effect of the copper concentration was studied in the interval between 1.68 × 10$^{-3}$ and 4.20 × 10$^{-4}$ mol·L$^{-1}$ Cu$^{2+}$, that is, between 0.11 and 0.027 g·L$^{-1}$ Cu$^{2+}$; Figure 7 shows the results. Figure 8 is a graph corresponding to the experiment performed at 1.68 × 10$^{-3}$ M Cu$^{2+}$, to show the

appearance of two stages, as in the other experiments; this does not appear in Figure 7. Table 3 includes the experimental constants for these experiments.

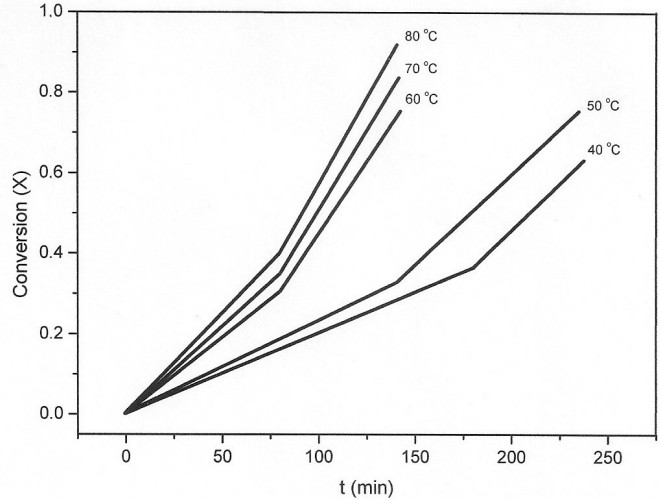

**Figure 4.** Effect of temperature on thiosulfate transformation: $[S_2O_3{}^{2-}]_{initial}$, 1 g·L$^{-1}$; $[Cu^{2+}]_{initial}$, 0.053 g·L$^{-1}$; pH$_{constant}$, 5; O$_2$ pressure, 1 atm.

**Table 2.** Effect of temperature on thiosulfate transformation.

| Temp. (°C) | 1000/T (K$^{-1}$) | $k_{exp(1)}$ (min$^{-1}$) | $k_{exp(1)}/[O_2]$ (min$^{-1}$/mol·m$^{-3}$) | Ln($k_{exp(1)}/[O_2]$) | $k_{exp(2)}$ (min$^{-1}$) | $k_{exp(2)}/[O_2]$ (min$^{-1}$/mol·m$^{-3}$) | Ln($k_{exp(2)}/[O_2]$) |
|---|---|---|---|---|---|---|---|
| 40 | 3.195 | 0.0020 | $2.08 \times 10^{-3}$ | $-6.177$ | 0.0045 | $4.67 \times 10^{-3}$ | $-5.366$ |
| 50 | 3.096 | 0.0022 | $2.65 \times 10^{-3}$ | $-5.933$ | 0.0057 | $6.87 \times 10^{-3}$ | $-4.981$ |
| 60 | 3.003 | 0.0038 | $5.35 \times 10^{-3}$ | $-5.231$ | 0.0075 | 0.0106 | $-4.550$ |
| 70 | 2.915 | 0.0042 | $7.24 \times 10^{-3}$ | $-4.928$ | 0.0080 | 0.0138 | $-4.280$ |
| 80 | 2.833 | 0.0050 | 0.0116 | $-4.457$ | 0.0086 | 0.020 | $-3.912$ |

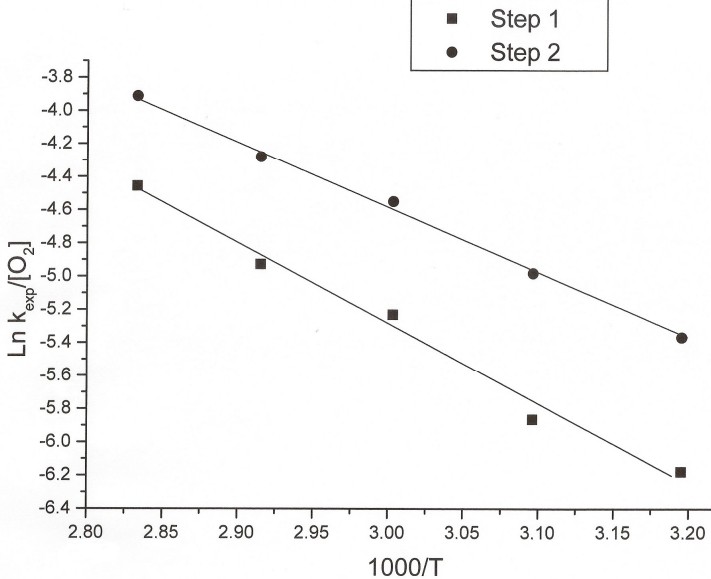

**Figure 5.** Apparent activation energy (E$_a$) for thiosulfate transformation (first step): 41 kJ·mol$^{-1}$, and (second step): 33 kJ·mol$^{-1}$.

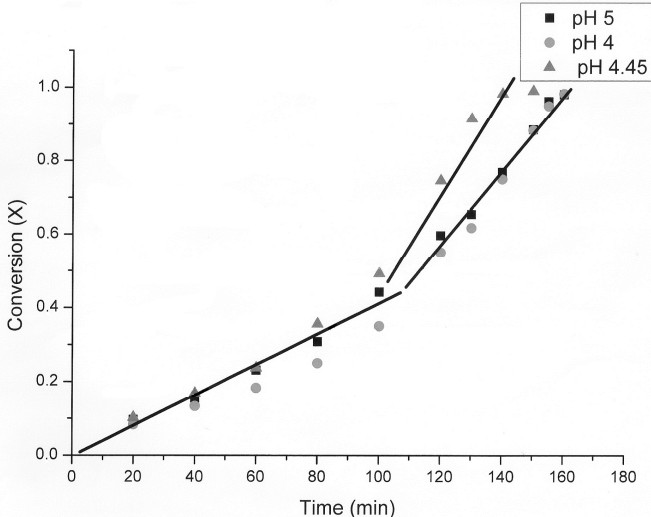

**Figure 6.** Effect of $H_3O^+$ concentration on thiosulfate transformation: $[S_2O_3^{2-}]$ $_{initial}$, 1 $g \cdot L^{-1}$; $[Cu^{2+}]$ $_{initial}$, 0.053 $g \cdot L^{-1}$; 60 °C; $O_2$ pressure, 1 atm.

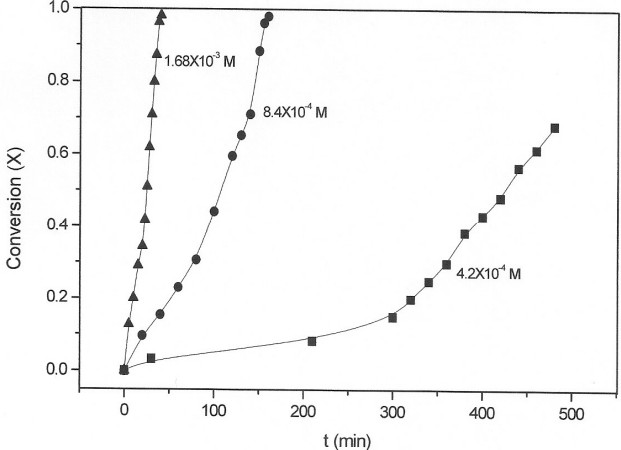

**Figure 7.** Effect of initial $Cu^{2+}$ concentration on thiosulfate transformation: $[S_2O_3^{2-}]_{initial}$, 1 $g \cdot L^{-1}$; $pH_{constant}$, 5; 60 °C; $O_2$ pressure, 1 atm.

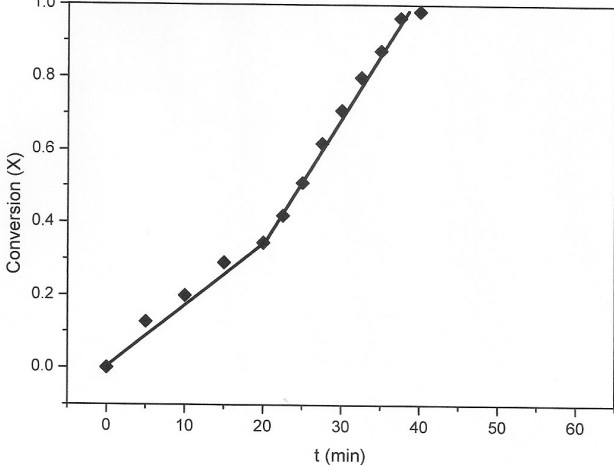

**Figure 8.** Effect of initial $Cu^{2+}$ concentration on thiosulfate transformation: Experiment performed at $1.68 \times 10^{-3}$ $mol \cdot L^{-1}$ $Cu^{2+}$ (0.107 $g \cdot L^{-1}$).

**Table 3.** Effect of copper concentration: values of the experimental constants.

| [Cu$^{2+}$] (mol·L$^{-1}$) | $k_{exp(1)}$ (min$^{-1}$) | $k_{exp(2)}$ (min$^{-1}$) |
|---|---|---|
| $4.2 \times 10^{-4}$ | 0.00044 | 0.0030 |
| $8.4 \times 10^{-4}$ | 0.0030 | 0.011 |
| $1.68 \times 10^{-3}$ | 0.017 | 0.034 |

Figure 9 is a graph of log $k_{exp}$ versus log initial [Cu$^{2+}$]. Apparent reaction orders of 2.5 for the first step and 1.75 for the second step were obtained. These values show the important effect that the initial concentration of copper (II) has on the rate of transformation of thiosulfates in the interval of copper concentrations from $4.2 \times 10^{-4}$ to $1.68 \times 10^{-3}$ mol·L$^{-1}$.

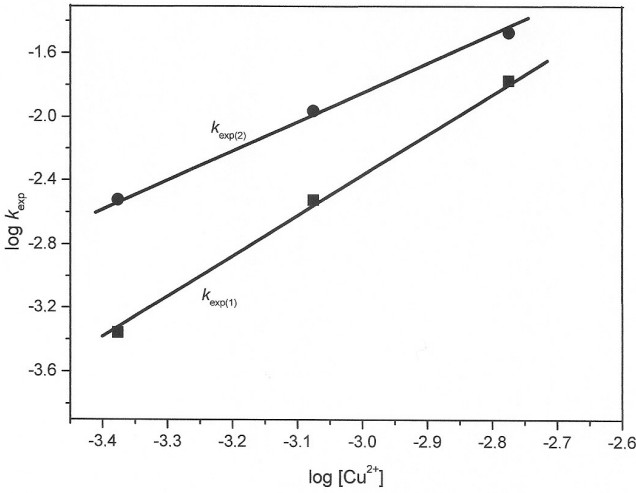

**Figure 9.** Effect of the initial Cu$^{2+}$ concentration on thiosulfate transformation: order of the reaction.

The effect of the initial concentration of thiosulfates on their transformation rate was determined for the interval $4.5 \times 10^{-3}$–$9 \times 10^{-3}$ mol·L$^{-1}$ (0.50–1 g·L$^{-1}$); Figure 10 includes the results. As the reaction rate decreases, as the initial thiosulfate concentration increases. When the molar ratio S$_2$O$_3^{2-}$/Cu$^{2+}$ is ≤8, only one $k_{exp}$ value was obtained; for values of the molar ratio >10, two slopes appeared. Table 3 includes values of the log of the initial thiosulfate concentration and the log of the different values of $k_{exp}$.

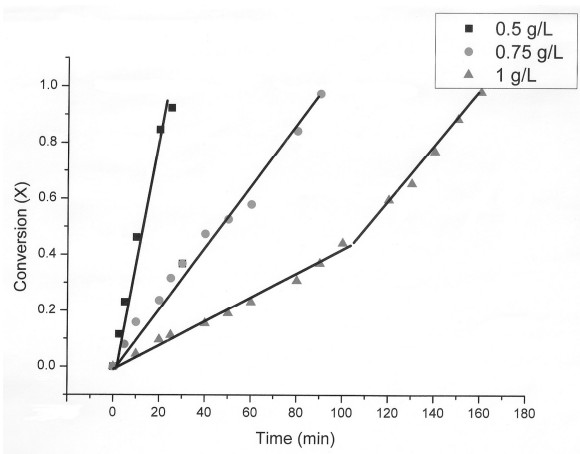

**Figure 10.** Effect of initial S$_2$O$_3^{2-}$ concentration on thiosulfate transformation: [Cu$^{2+}$]$_{initial}$, 0.053 g·L$^{-1}$; pH$_{constant}$, 5; 60 °C; O$_2$ pressure, 1 atm.

Table 4, lines 1 and 2, contain the $k_{exp}$ values corresponding to the experiments in which 0.5 and 0.75 g·L$^{-1}$ of thiosulfate were used; line 3 includes the average value of $k_{exp}$ obtained in the 1 g·L$^{-1}$ thiosulfate experiment (under these conditions, a change of slope was detected). Lines 4 and 5 correspond to the $k_{exp}$ values of the first and second stage, respectively, of the experiment with 1 g·L$^{-1}$ of thiosulfates. The apparent reaction order, considering the first three lines, is −2.8; if the experiment at 1 g·L$^{-1}$ is considered in two steps, the apparent order of the first step is −3.4 and the apparent order of the second is −2.6. When the thiosulfate concentration was increased, the reaction rate decreased significantly, because of the increases in the thiosulfate/Cu(II) ratio.

**Table 4.** Reaction order with respect to the initial thiosulfate concentration.

| Line | $[S_2O_3{}^{2-}]$ (mol·L$^{-1}$) | log $[S_2O_3{}^{2-}]$ (mol·L$^{-1}$) | $k_{exp}$ (min$^{-1}$) | log $k_{exp}$ (min$^{-1}$) |
|---|---|---|---|---|
| 1 | $4.46 \times 10^{-3}$ | −2.351 | 0.0410 | −1.387 |
| 2 | $6.70 \times 10^{-3}$ | −2.174 | 0.0098 | −2.009 |
| 3 | $8.92 \times 10^{-3}$ | −2.050 | 0.0060 | −2.222 |
| 4 | $8.92 \times 10^{-3}$ | −2.050 | 0.0040 | −2.398 |
| 5 | $8.92 \times 10^{-3}$ | −2.050 | 0.0070 | −2.155 |

An additional experiment was carried out by decreasing the $S_2O_3{}^{2-}/Cu^{2+}$ ratio, using $3.6 \times 10^{-2}$ mol·L$^{-1}$ thiosulfate (4 g·L$^{-1}$ thiosulfate) and an initial $Cu^{2+}$ concentration of $2.5 \times 10^{-3}$ mol·L$^{-1}$ (initial thiosulfate/copper molar ratio of 14.2). The behavior was quite similar to that shown in Figure 8 (initial copper concentration $4.2 \times 10^{-4}$ mol·L$^{-1}$). The rate values obtained were: from 0 to 270 min, $k_{exp1} = 0.0020$ and $k_{exp2} = 0.0083$.

The effect of the partial pressure of oxygen was determined in the interval between 0 and 1 atm pressure; Figure 11 shows the results. The reaction rate increases as the partial pressure of oxygen increases. In the experiment without oxygen, a conversion of 0.33 was achieved after 480 min reaction time (First step: $k_{exp} = 0.0006$ min$^{-1}$). When air was used (partial pressure of oxygen = 0.2), a conversion of 0.50 was achieved at the same time (First step: $k_{exp} = 0.0010$ min$^{-1}$). Finally, with oxygen at a partial pressure of 1 atm, a conversion of 0.98 was obtained in 160 min ($k_{exp} = 0.0040$ min$^{-1}$, first step; and $k_{exp} = 0.0070$ min$^{-1}$, second step). Only in the last experiment was the degradation reaction almost completed. Consequently, an average value of reaction order near to 1 was obtained with respect to the partial pressure of oxygen. The reaction rate was multiplied by 4 when oxygen was used instead of air, i.e., approximately the same ratio of the oxygen partial pressure in air or in pure oxygen.

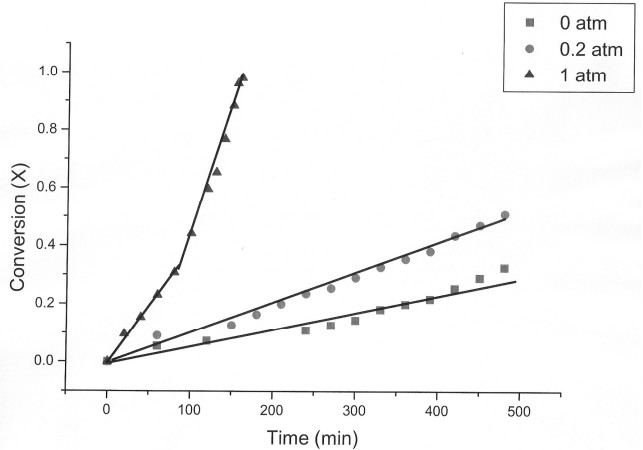

**Figure 11.** Effect of the partial pressure of oxygen on thiosulfate transformation: $[S_2O_3{}^{2-}]$, 1 g·L$^{-1}$; $[Cu^{2+}]_{initial}$, 0.053 g·L$^{-1}$; pH$_{constant}$, 5; 60 °C.

The transformation of thiosulfate to sulfate using oxygen and copper (II) in dilute solution (25–100 ppm), at pH 5 and 60 °C, has potential applications in treating effluents for thiosulfate degradation. The small amounts of copper sulfide generated can be oxidized to copper sulfate and recycled to the next cycle of thiosulfate degradation.

## 4. Conclusions

(1) In the degradation of thiosulfate by oxygen and small amounts of $Cu^{2+}$ at pH between 4 and 5, a change in the slope of the graph of reaction rate ($k_{exp}$) versus time was observed. This coincided with a precipitate of CuS being formed, and the reaction rate increased from this point. This implies a catalytic effect of contact with CuS on the thiosulfate degradation.

(2) The reaction produced in the zone where the change in the slope occurs could be as follows:

$$S_2O_3^{2-}{}_{(aq)} + Cu^{2+}{}_{(aq)} + 3H_2O \Rightarrow SO_4^{2-}{}_{(aq)} + CuS_{(s)} + 2H_3O^+{}_{(aq)}.$$

Analysis of the final liquids carried out by chromatography confirmed that the thiosulfates were oxidized to sulfates under the employed experimental conditions. The amount of thiosulfate remaining in solution at the end of the process was less than 20 mg·L$^{-1}$.

(3) In the kinetic study of thiosulfate degradation, two apparent activation energies were obtained: 41 kJ·mol$^{-1}$ (in the first step, until a conversion of nearly 0.35) and 33 kJ·mol$^{-1}$ (in the second step after CuS formation). The effect of temperature on the degradation process using oxygen and Cu (II) at the ppm level is less than that observed when using larger amounts of Cu (II) without oxygen (98 kJ·mol$^{-1}$).

(4) For a concentration of Cu (II) varying between $1.68 \times 10^{-3}$ and $4.20 \times 10^{-4}$ mol·L$^{-1}$ $Cu^{2+}$ (107–27 ppm Cu in solution), the reaction rate increased with the Cu (II) concentration, giving an apparent reaction order near to 2 (2.5 for the first step and 1.75 for the second step).

(5) For a concentration of thiosulfate varying between $4.5 \times 10^{-3}$ and $9 \times 10^{-3}$ mol·L$^{-1}$ $S_2O_3^{2-}$, only one reaction rate (slope) was detected for molar ratios thiosulfate/Cu $\leq 8$. Two rates (slopes) were detected for values of this ratio higher than 10. Apparent reaction orders of $-3.4$ (first step) and $-2.6$ (second step) were obtained, indicating that, when the thiosulfate concentration increases, the reaction rate decreases significantly, because of the increases in the thiosulfate/Cu(II) ratio.

(6) The effect of the partial pressure of oxygen was determined in the interval between 0 and 1 atm pressure. The reaction rate increases as the partial pressure of oxygen increases. An average apparent reaction order near to 1 was obtained.

(7) The transformation of thiosulfate to sulfate by oxygen and a dilute copper (II) solution (25–100 ppm), at pH 5 and 60 °C, has potential applications in treating effluents for thiosulfate degradation. The CuS generated during the process can be oxidized and recycled to the next cycle of thiosulfate degradation.

**Author Contributions:** A.R.V., M.C.C. and F.P.C. conceived and designed the experiments; J.M.G.L. performed the experiments; A.R.V., M.C.C., J.M.G.L. and F.P.C. analyzed the data; and A.R.V. and M.C.C. wrote the paper.

**Acknowledgments:** The authors wish to thank the *Centres Científics i Tecnològics* of the Universitat de Barcelona for their assistance with this work.

**Conflicts of Interest:** The authors declare no conflict of interest.

## List of Symbols

| | |
|---|---|
| Conversion | *X* (dimensionless) |
| $k_{exp}$ | reaction rate (min$^{-1}$) |
| Activation energy | Ea (kJ·mol$^{-1}$) |

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
