# Peer review of "Oxidation of Thiosulfate with Oxygen Using Copper (II) as a Catalyst"

_metals, doi:10.3390/met9040387_

Round 1

Reviewer 1 Report

What is the difference between this work and work done in 2009? I find this article in same theory, and couldn’t find significant difference with previous work. 

Line 81. The same logic was in previous work of authors, avoiding formation of tetrationate.

Line 191. Conversion word

Author Response

Reviewer 1

First of all, thank you very much for your comments and suggestions. Here you find enclosed the changes made in the paper.

1)      We have improved the paper from the comments of the reviewers in all of items. Particularly, two equations, a table and two references have been included, jointly with additional comments in the “results discussion” item. Four figures have been improved with respect to the first manuscript

2)      The difference between this work (degradation using oxygen and copper (II) at ppm level) and that published in Hydrometallurgy in 2009 (degradation using copper (II) in an amount near to the stoichiometric with respect to the amount of thiosulfate) has been included in the text at the end of page 4 and in page 5.

3)      English has been revised by an English native professor of the University of Barcelona (Language Services of our university).

Reviewer 2 Report

The authors investigated the removal of thiosulphate by oxidation in aqueous solution using Cu(II) as catalyst. This is a process of interest in view of the expected increase in thiosulphase in waste streams. Many experiments have been performed yielding reaction rates, activation energies and reaction orders. The results are useful for designing a process to remove the thiosulphate. From scientific point of view a deeper analysis of the reaction mechanism to explain the observed kinetics and also an overall kinetic model is unfortunately missing. Particularly the influence of oxygen is not explained at all (at least some speculation should be included). Maybe O2 reacts with a precursor of CuS, thus effectively in-situ regenerating the Cu. Also the highly negative reaction order of thiosulphase (about -3) is rather peculiar and needs further discussion. Another issue which is only briefly mentioned at the end is the reaction of Cu catalyst to CuS, which needs to be regenerated, which is not easy. Some more attention on that aspect should be included. If the amount of CuS is significant, you may even not call the Cu a catalyst.

Minor comments:

Line 30: the last part of the sentence starting with ’In metallurgy’ is not correct (‘it is difficult the recovery’ is not correct English).

Line 41: Consider changing ‘An alternative includes’ to ‘An alternative process includes’

Line 36: What is meant here with ‘of interference from’ ? Maybe ‘interaction with’ is meant here.

Table 1: Please specifiy the reaction rate equation used. E.g. r = k_exp [S2O3^2-]. Also it should be specificied how the O2 concentration was obtained. Was it measured (if yes, how ?) or did you use a solubility equation (please give this equation). Please also add a list of symbols used (including units).

Author Response

Reviewer 2

First of all, thank you very much for your comments and suggestions. Here you find enclosed the changes made in the paper.

1)      We have improved the paper from the comments of the reviewers in all of items. Particularly, two equations, a table and two references have been included, jointly with additional comments in the “results discussion” item, and concretely in relation to the effects of thiosulfate concentration and oxygen partial pressure. Four figures have been improved with respect to the first manuscript

Minor comments:

2)      Lines 30, 36 and 41: corrected

3)      The expression used for the activation energy calculation has been included in the text

4)      The information for obtaining the oxygen concentration at different temperatures have been indicated in the text as a reference

5)      Additional comments about the morphology and size of the copper sulfide precipitated have been included. The amount of this precipitate generated is very low (the amount of copper used is about 50 ppm).

6)      A list of symbols have been added at the end of the paper

7)      English has been revised by an English native professor of the University of Barcelona (Language Services of our university).

Reviewer 3 Report

Overall, the research method and analysis  of results are well-documented and make sense. The authors were presented very well  introduction literature overview, too. The subject and the methodology applied seem to be justified and deserve publication. The topic is very importent becouse in world is increase problems  recovery Ag and Au.  The literature is properly cited.
However  the figures presented in the article are not good quality, a esppecially figures 2, 3 and 4 - and the authors must improved their quality. I have a question of the methodology  determination concentration of Cu(II) - what apparatus was used, what methodology and what is the measurement error.

Author Response

Reviewer 3

First of all, thank you very much for your comments and suggestions. Here you find enclosed the changes made in the paper.

1)      We have improved the paper from the comments of the reviewers in all of items. Particularly, two equations, a table and two references have been included, jointly with additional comments in the “results discussion” item. Four figures have been improved with respect to the first manuscript; now are of better quality

Additional comments:

2)      Copper (II) concentration was not determined. In all cases the conversion process was followed by determining the thiosulfate concentration at each time.

3)      English has been revised by an English native professor of the University of Barcelona (Language Services of our university).

Round 2

Reviewer 1 Report

Authors edit satisfies my recommendations.